# Assessing the Impact of Man–Made Ponds on Soil Erosion and Sediment Transport in Limnological Basins

**Mario J. Al Sayah** [1,2,3], **Rachid Nedjai** [3], **Konstantinos Kaffas** [4], **Chadi Abdallah** [1,*] and **Michel Khouri** [2]

1 National Council for Scientific Research, Remote Sensing Center, Beirut 11-8281, Lebanon; mario.alsayah@gmail.com
2 Centre de Recherches en Sciences et Ingénierie, Lebanese University Faculty of Engineering II, Roumieh 1205, Lebanon; mkhuri@ul.edu.lb
3 Centre d'Études et de Développement des Territoires et de l'Environnement, Université d'Orléans, 45100 Orléans, France; rachid.nedjai@univ-orleans.fr
4 Faculty of Science and Technology, Free University of Bozen, 39100 Bozen, Italy; Konstantinos.Kaffas@unibz.it or kostaskaffas@gmail.com
* Correspondence: chadi@cnrs.edu.lb; Tel.: +961-3-534-436 or +961-4-409846; Fax: +961-4-409845

**Abstract:** The impact of ponds on basins has recently started to receive its well-deserved scientific attention. In this study, pond-induced impacts on soil erosion and sediment transport were investigated at the scale of the French Claise basin. In order to determine erosion and sediment transport patterns of the Claise, the Coordination of Information on the Environment (CORINE) erosion and Soil and Water Assessment Tool (SWAT) models were used. The impact of ponds on the studied processes was revealed by means of land cover change scenarios, using ponded versus pondless inputs. Results show that under current conditions (pond presence), 12.48% of the basin corresponds to no-erosion risk zones (attributed to the dense pond network), while 65.66% corresponds to low-erosion risk, 21.68% to moderate-erosion risk, and only 0.18% to high-erosion risk zones. The SWAT model revealed that ponded sub-basins correspond to low sediment yields areas, in contrast to the pondless sub-basins, which yield appreciably higher erosion rates. Under the alternative pondless scenario, erosion risks shifted to 1.12%, 0.52%, 76.8%, and 21.56% for no, low, moderate, and high-erosion risks, respectively, while the sediment transport pattern completely shifted to higher sediment yield zones. This approach solidifies ponds as powerful human-induced modifications to hydro/sedimentary processes.

**Keywords:** soil erosion; sediment transport; SWAT; CORINE erosion model; ponds; Brenne; limnology; land cover change

## 1. Introduction

Understanding the impact of land occupation (land use/cover) on basin processes, such as rainfall-runoff and soil erosion, is an integral part of land and water management-oriented decisions [1,2]. The processes of soil erosion and sediment transport take part in van Rijn's (1993) [3] sedimentary cycle, and often are the main causes of soil loss in basins [4]. Although these processes are of natural origin [5], the interaction between climate, soil, topography, land use, and land cover significantly influences erosion rates and sediment loads [6,7]. Soil loss due to these processes is a frequent problem that hydrologists, land planners, and basin managers will need to contend with [8]. Accordingly, soil erosion quantification in erosion-prone areas, with the highest accuracy possible, provides a complete

knowledge of soil loss hotspots and allows prioritized treatment measures for supporting land processes in the concerned basin [9]. Subsequently, the reliable assessment and representation of sediment yields—which depend on the cascading effect of soil erosion—allows an in-depth understanding of the soil erosion-sediment link at the basin scale [9].

For representing both processes, several models have been developed and used extensively to replace the conventional assessment methods, i.e., the Water Erosion Prediction Project (WEPP) [10,11], the Universal Soil Loss Equation (USLE) [12], the Modified Universal Soil Loss Equation (MUSLE) [13], the Revised Universal Soil Loss Equation (RUSLE) [14,15] and the Soil and Water Assessment Tool (SWAT) [16]. SWAT is one of the most widely used basin models. It has been applied extensively in modeling the impact of land occupation changes, under different scenarios and different contexts [17,18]. The widespread use of SWAT can be justified by its sensitivity and flexibility towards the land occupation input [19], its adaptability to different contexts—even to those with data scarcity [20]—its simple data requirements and ease of computation [21], as well as the straightforward calibration through its stand-alone SWATCUP interface [22].

Despite abundant research regarding the impact of land occupation on soil loss, few studies focus on the particular case of small water bodies and their effect as a land occupation class [23]. Small water bodies, like ponds and wetlands, are considered as the most amplified form of human-induced modifications to the hydro-sedimentological system of basins [24]. Ponds represent a total of over 90% of global standing water bodies, 30% of global standing waters by surface area [25], and form the most widespread aquatic habitat dominating the continental standing waters in Europe [26]. Despite their well-documented significance [27], abundant numbers, and increasing proliferation [28], ponds have not received considerable scientific attention compared to rivers and lakes [26]. It is worth mentioning that research regarding ponds in Europe has tripled in the last decade [29], where results showed that ponds contribute significantly to several basin related processes [30]. Examples of these processes are sediment interception [31], removal of pollutants for river protection [32], nutrient recycling [33], greenhouse gas emission [34], regulation of hydrological flows [35], biogeochemistry [30], and climate [36]. In addition to their environmental role, ponds have a well-known value for housing and sustaining biodiversity, supporting livelihoods, local economies, and taking part in the socio-cultural heritage of the settings in which they are located [37].

Under the hydro-sedimentological scope, particularly, ponds have shown to retain as much as 90% of sediments transported in basins [38]. Consequently, ponds have been heavily blamed for rupturing the ecological and sedimentary continuum of the basins to which they belong [39]. The disruptive effect of ponds is due to the increase of residence time of waters, resulting in a decline in the temporal variation of the main discharge [40]. Accordingly, the deceleration of overland flow allows suspended particles to settle under the effect of their weight, causing a reduction in the amount of sediments entrained by water, making ponds sediment sinks [40]. However, this effect strongly depends on their position in the basin, their depth, volume, slope [41], as well as the surrounding land occupation [42]. Winfield Fairchild and Velinsky (2006) [43] showed that ponds located upstream of rivers—considering their sediment retention capacity—are capable of creating a state of imbalance in the geochemical and hydro-sedimentary status of the underlying rivers. Consequently, the Directive Cadre sur l'Eau (DCE) [44] stresses the need to assess the impact of hydromorphological elements that are capable of influencing hydrologic pathways, river morphology, width, and continuity.

Beyond the contribution of isolated ponds, connected networks of ponds were found to contribute to basin processes at higher rates than lakes or even rivers [45,46]. Particularly in France, ponds are mainly concentrated in three regions: the Sologne region, Brenne (Central France), and Dombes (Eastern France). In response to DCE recommendations, this study aims to assess the impact of man–made ponds on soil erosion and sediment transport, at the scale of the Indre portion of the Claise basin. This part corresponds to the Brenne Natural Regional Park that houses 4500 waterbodies (ponds, marshes, and small water surfaces), 2179 of which are located in Claise, being part of an interconnected network. To evaluate erosion risks in the Claise, the Coordination of Information on the Environment

CORINE (1992) Erosion Risk model will be used, since it presents a simplification of the reliable USLE model [1] and given that no-erosion field data, for the Claise, were available for the study. SWAT is employed to assess the impact of ponds on the Claise's hydro-sedimentary regime. This choice is owed to SWAT's ability to simulate the physical processes that occur in ponds, which in turn allows an accurate representation of pond containing basins [38]. This is achieved through the SWAT Pond (.pnd) input file that makes SWAT one of the few hydrological models having an input for ponds [47]. Related studies have assessed the behavior of ponds using SWAT [38,48] and highlighted the efficiency of SWAT on this part.

The impact of the Claise's ponds on erosion and sediment transport will be assessed by testing alternative scenarios, where the land occupation input for both models will be simulated with and without ponds. By this approach, a quantification of the pond impact can be obtained. The presented work serves as a decision-oriented tool for basins similar to the Claise, where pond proliferation has been halted until a proper understanding of their effect is established. In addition, analysis of soil erosion risks and sediment transport is useful for conservation measures that aim towards prolonging the useful life of these small water bodies or for ceasing their proliferation.

## 2. Materials and Methods

### 2.1. Study Area

The Brenne portion of Claise basin (Figure 1) covers an area of 707 km$^2$ and is one of the three entities housing the 4500 ponds of the Brenne Natural Regional Park, which are mostly grouped in the form of interconnected chains [36]. Within the Claise, 2179 of the Brenne waterbodies, along with the Five Bonds channel (Blizon), are the main contributors [49] to the 87.6 km long Claise River having an average discharge of 4.5 m$^3$/s [50]. The basin is mostly dominated by a degraded oceanic climate with a mean temperature of 11 °C and average annual precipitation of 700 mm [51]. As to its topographical profile, the basin is considered to be a flat plain, with 99% of its surface falling into the 0%–5% slope class, while its altitude varies between 76 m and 181 m. The six poorly-permeable soil classes of the basin are mostly dominated by Luvisols [36,52]. At a combined state, the Claise's challenging pedology, flat topographical setting, and quasi–impermeable lithology have resulted in the stagnation of incoming water leading to the formation of natural ponds [53]. According to Bennarrous (2009) [37], however, these ponds are not only a product of natural processes but also an anthropogenic adaptation to a poorly drained domain and a source of economic livelihood (aquaculture) in an environment of limited productivity. As a result of intensive pond proliferation throughout time, the basin has acquired a particular hydrographic network characterized by an abundance of different kinds of water bodies. Despite its richness, nonetheless, the hydrographic network of the basin is randomly organized and presents severe fragmentation [54]. In contrast to the evolving pond proliferation, the land occupation setting of the Claise has been relatively unchanged for the last 19 years, mainly displaying a dominance of an interlocked mosaic of forests, grasslands, and agricultural areas [54].

### 2.2. The SWAT Model: Basins and Impoundments

SWAT is one of the few hydrological models that have an input for ponds [47], which makes it an ideal tool for this study. SWAT is a semi-physical deterministic distributed and continuous hydrologic model that functions at a daily time step with options for sub-hourly routing, as well [55]. It is a quite complex model with significant input data demands. However, the basic components can be readily obtained and are relatively simple [56]. Essential inputs consist of weather data, digital elevation model (DEM), soil, and land occupation maps [57]. Based on these inputs, SWAT divides the basin into sub-basins, which further divides into smaller hydrological response units (HRUs) [58]. HRUs are defined as units with homogeneous land occupation, topography, and pedological properties [59]. These are generated in order to lump somewhat similar areas scattered through the basin into a single unit, simplifying the model's run by avoiding unpractical simulations while accounting for

the diversity of different factors in the basin [60]. Within SWAT, ponds are defined as waterbodies, being an integral part of a sub-basin's hydrological network; they are capable of intercepting surface runoff [60], thereby modifying the hydro-sedimentary behavior of the basin. Since this work targets the erosion/sedimentary behavior of the SWAT model, only their related equations will be presented. Further details can be found in Neitsch et al. (2011) [47].

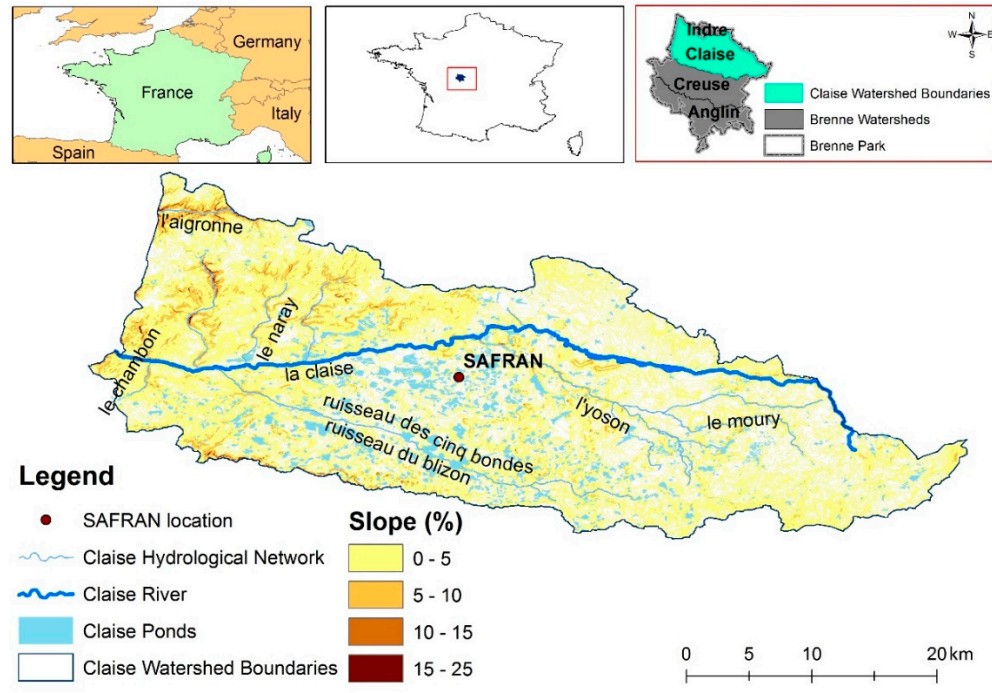

**Figure 1.** Study area.

Soil erosion is calculated based on the Modified Universal Soil Loss Equation (MUSLE) using the following formula [55]:

$$sed = 11.8 \times (Q_{surf} \cdot q_{peak} \cdot area_{hru})^{0.56} \times K_{USLE} \times C_{USLE} \times P_{USLE} \times LS_{USLE} \times CFRG \tag{1}$$

where *sed* is the HRU sediment yield (t); $Q_{surf}$ is the surface runoff volume (mm); $q_{peak}$ is the peak runoff rate (m³/s); $area_{hru}$ is the area of the HRU (ha); $K_{USLE}$ is the Universal Soil Loss Equation (USLE) soil erodibility factor; $C_{USLE}$ is the USLE cover and management factor; $P_{USLE}$ is the USLE support practice factor; $LS_{USLE}$ is the USLE topographic factor; and $CFRG$ is the coarse fragment factor.

It should be noted that MUSLE virtually calculates the sediment yields due to soil erosion that reach the main streams of the sub-basins in a unit of time. This is the reason why sediment delivery ratios are not required, contrarily to USLE and RUSLE [61].

The mass balance equation of sediment in ponds is described by the following formula [47]:

$$sed_{wb} = sed_{wbi} + sed_{flowin} - sed_{stl} - sed_{flowout} \tag{2}$$

where $sed_{wb}$ is the amount of sediment at the end of the day; $sed_{wbi}$ is the amount of sediment at the day's beginning; $sed_{flowin}$ is the amount of sediment provided from inflows; $sed_{stl}$ is the amount of settled sediments; and $sed_{flowout}$ is the amount of sediment transported as outflow. All components are expressed in metric tons.

*2.3. The CORINE Erosion Model*

The CORINE erosion model is a simplification of the USLE model. In the CORINE model, erosion risks are classified on a scale of 0–3, with 0 corresponding to the no-erosion class, 1 to low erosion risks,

2 to moderate erosion risks, and 3 to high-erosion risks [62]. For the estimation of erosion using the CORINE model, parameters such as soil erodibility, climate erosivity, topography (slope), and LU/LC (vegetation cover) are required [61]. Each parameter, in turn, consists of several sub-factors, and is classified according to the CORINE model into respective indices (Figure 2). Once established, the soil erodibility index is combined with climate erosivity and slope, to yield the potential soil erosion risk map. Subsequently, the potential soil erosion risk map indices (0–3) are crossed with those of the vegetation cover to yield the actual soil erosion map.

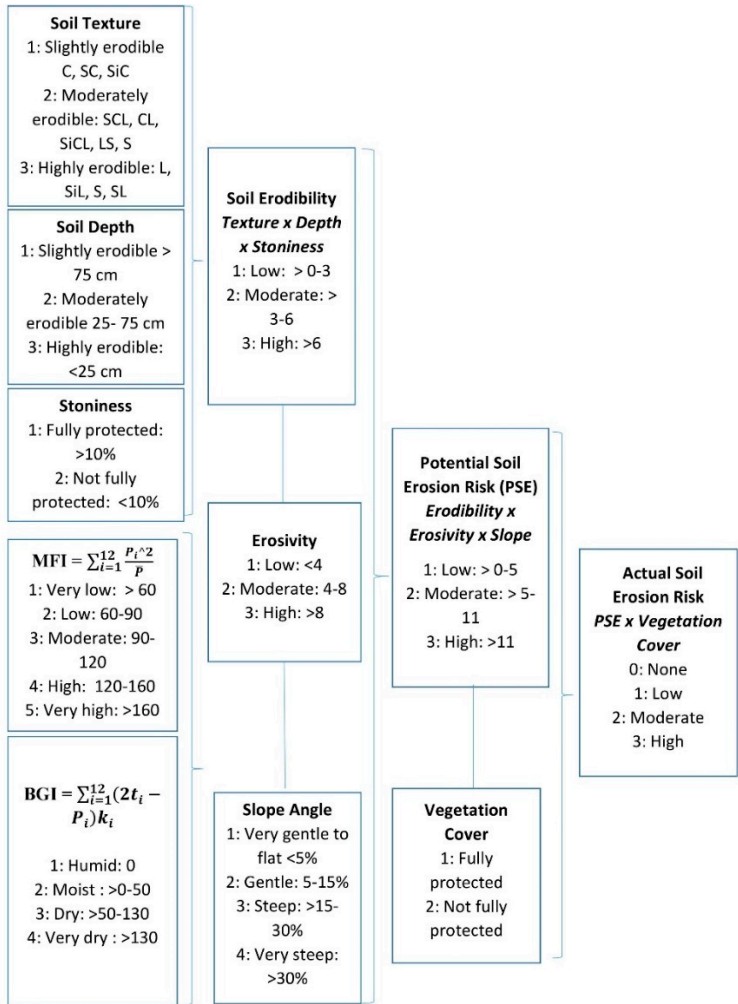

**Figure 2.** CORINE model framework, MFI: Modified Fournier Index [63]; $P_i$ is the total precipitation in month i and $\overline{P}$ is the mean annual total precipitation; BGI: Bagnouls–Gaussen Index [64]; $t_i$ the mean temperature for the month i; $P_i$ the total precipitation in month i and $k_i$ the proportion of the month i in which $2t_i - P_i > 0$ [62].

## 2.4. Input Data and Adaptation to the SWAT and CORINE Models

The input data used in this study, along with a short description, are summarized in Table 1.

**Table 1.** CORINE erosion model and SWAT input data.

| Input Data | Source | Date [†] | Description |
|---|---|---|---|
| Topography—DEM | Institut Géographique National (IGN)—France | 2010 | 25 m resolution |
| Soil map | Harmonized World Soil Database (HWSD) [52] | 2008 | 30 arc-second raster database |
| Weather data | Weatherlink Pro 2 weather stations, source: R. Nedjai (Dynétangs) | 2016–2018 | Daily time step |
| | Système d'Analyse Fournissant des Renseignements Adaptés à la Nivologie (SAFRAN) model (Durand et al., 1993) | 1970–2018 | |
| Land Use/Land Cover map | Digitized from aerial photography and cross-checked against ancillary maps | 2018 | Ortho-rectified aerial photography-Resolution: 0.5 m verified against ancillary maps: CORINE 2012 land use/land cover maps |

[†] The correspondence of the Claise basin to a natural park renders any human induced modifications on soils minor. Further, the Claise's land occupation pattern is relatively stable. Therefore, climate is the only variable factor in the study area. Hence, the temporal difference of the utilized datasets does not cause temporal induced biases.

ArcSWAT version 2012.10_3.19 was applied in an ArcGIS 10.3 (Environmental Systems Research Institute, Inc. (Esri), 380 New York Street, Redlands, California, USA) environment to perform simulations at a daily time step for the period 1970–2018. The first seven years of the simulation, from 1970 to 1976, were used for the model's warm up. The calibration and validation phases of the model were carried out by means of comparison between simulated discharges and measured discharge data from the station L6202040–La Claise au Grand–Pressigny (Pont de Fer), obtained from Eau France Banque HYDRO–Ministère de l'Ecologie, de l'Energie, du Développement Durable et de l'Aménagement du Territoire (MEEDDAT)/Direction Générale de la Prévention des Risques (DGPR)/Service des Risques Naturels et Hydrauliques (SRNH). Due to the unavailability of sediment records for this study, only a hydrologic calibration was performed. However, surface runoff and stream flow—regardless of the approach used to model the sedimentary cycle—are the driving forces of runoff and streambed erosion, as well as of overland and stream sediment transport [65]. Hence, one might say that the calibration of the sediment part of the study might be implicit, but nonetheless robust. According to Hallouz et al. (2018) [66], a calibrated hydrologic model gives a certain degree of reliability to the sedimentary output. However, the availability of sediment data is often a constraint in relative studies, as measurements, either in the form of gross or net erosion or in the form of total sediment discharge, in streams, are often nonexistent. Even in cases where such data is available, it is in the form of sparse discrete measurements and rarely in the form of continuous sediment graphs, where they could be used for a robust calibration. In the case of the Claise, this would be a very challenging task, given the large number of ponds. For this reason, SWAT was calibrated according to the calibration scheme of Jalowska and Yuan (2019) [38], where a complete representation of an impoundment effect (here, ponds) is ensured. Nonetheless, the absence of sediment records poses some short of limitation that needs to be considered by the decision-makers of the basin.

2.4.1. DEM: Topographic Effect and Use in Both Models

In the CORINE model, DEM is used to extract the topographic parameter by means of slope computation. The slope angles of the study area were obtained using the "slope" tool of ArcGIS. Subsequently, the slope raster was classified, with respect to the CORINE's model indices, into (1) very gentle to flat (<5%), (2) gentle (5%–15%), (3) steep (15%–30%), and (4) very steep (>30%).

In SWAT, the DEM is used for the extraction of the topographic parameters and for representing the basin's physical parameters such as slopes, flow direction and accumulation, delineation of the hydrologic network, and basin partitioning [67]. After inputting the DEM into SWAT, the basin was

initially divided into 25 sub-basins using SWAT's watershed delineator. However, given the presence of a great number of waterbodies, and their impact on output accuracy, the number of sub-basins was increased to 35. The purpose of increasing the sub-basins' number was to increase the spatial resolution for a more detailed representation of the basins' processes [68]. Since the SWAT model allows only one pond per sub-basin [39] during the sub-basins' delineation, the largest number of ponds was included in each in order to ensure maximal representation of pond-induced processes. Further, all ponds within each sub-basin were lumped and the outlets of the sub-basins were chosen to coincide with those of the ponds in order to maximize their representation and account for their effect. The SWAT delineated hydrologic network was, in turn, verified against a pre-defined stream network in order to ensure and improve its accuracy.

### 2.4.2. Pedology: Adaptation to the Different Requirements of the Models Used

The pedological composition of the Claise was determined from the Harmonized World Soil Database (HWSD) [52]. For the CORINE model, soil texture was determined using the USDA textural triangle after inputting the respective percentages of sand, silt, and clay for each soil group obtained from the HWSD. Texture was then classified with respect to the CORINE indices. The parameters of depth and stoniness were treated and classified similarly. After obtaining the texture, depth, and stoniness sub-factors, the three layers were input into the "raster calculator" in analogy to Figure 2.

For the SWAT model, the hydrologic soil group of each class was assigned following the United States Department of Agriculture [69] Soil Conservation Service (SCS) soil survey. Bulk densities were computed using the Soil Water Characteristics software following Saxton and Rawl (2006) [70] equations that were shown to have adequate accuracy for bulk density computation by Al Sayah et al. (2019) [71]. Likewise, the available water capacity and saturated hydraulic conductivities were also computed using the same software. Organic carbon content was determined by multiplying the organic matter content derived from the HWSD by 0.58, since organic carbon forms around 58% of the soil's organic matter as a rule of thumb [72]. Sand, silt, and clay percentages, as well as the rock fragment contents were extracted from the HWSD. The USLE_K factor was computed following the formula presented in the SWAT documentation [55], while soil surface albedo was determined using the following formula [73]:

$$Soil\ albedo\ (0.3 - 2.8\ \mu m) = 0.069 \cdot (color\ value) - 0.114 \qquad (3)$$

After building the soil database, all parameters were reclassified in SWAT using user-adapted look-up tables for integration into the SWAT database.

### 2.4.3. Weather Data: Forcing on-Field Weather Data to a Meteorological Model

Weather data was obtained from two sources, on-field weather station data and Météo France's Système d'Analyse Fournissant des Renseignements Adaptés à la Nivologie (SAFRAN) model [74]. The SAFRAN model was used since it contains records for the period 1970–2018 while the weather stations, located next to a pond network, were used to account for the climatic regulating effect reported by Nedjai et al. (2018) [36]. SAFRAN was used in order to provide a large time span for the study. Moreover, the SAFRAN was validated by test-correlation with weather station data using Pearson's correlation coefficient "r" [75]. Prior to correlation, harmonization of both datasets was performed since the weather stations record parameters at the hourly time step, while SAFRAN simulates at the daily time step. Results of the correlation are presented in Table 2.

**Table 2.** Validation testing of the SAFRAN model against measured on field-data for revealing SAFRAN's validity for use.

| Weather Stations Parameters | SAFRAN Parameters | Parameter Label | Correlation, r |
|:---:|:---:|:---:|:---:|
| Temp Out | T_Q | Average temperature (°C) | 0.98 |
| Hi Temp | TSUP_H_Q | Maximal temperature (°C) | 0.98 |
| Low Temp | TINF_H_Q | Minimal temperature (°C) | 0.97 |
| Out Hum | HU_Q | Relative humidity (%) | 0.97 |
| Wind Speed | FF_Q | Wind speed (m/s) | 0.83 |
| Rain | PE_Q | Efficient rainfall (mm) | 0.64 |
| Solar Rad. | SSI_Q | Incoming solar radiation $(J/cm^2)$ | 0.98 |

For the CORINE model, the rainfall and temperature parameters are used for computation of the MFI and BGI indices following their respective formulas (Figure 2); these indices were then used to calculate erosivity following the workflow of Figure 2. In SWAT, the weather database was built using the WGN maker macro-tool [76] and input into the SWAT database.

### 2.4.4. Land Use and Land Cover: A Particularly Rich Natural Limnological Setting

The land occupation map of the Claise was obtained by on-screen digitizing of aerial photography at 0.5 m resolution. Such fine scale was used in order to ensure a detailed representation of the study area, particularly to account as accurately as possible for the scale of ponds.

For the CORINE model, the obtained land occupation map was reclassified into fully protected (forest, permanent pasture, and scrublands) and not-fully protected (cultivated or bare land) areas as demonstrated in Table 3.

**Table 3.** Numerical distribution of the land occupation setting of the Claise basin.

| Land Occupation Class | Area ($km^2$) | Percentage (%) | CORINE Vegetation Cover Index |
|:---:|:---:|:---:|:---:|
| Clear broad—leaved forest | 2.21 | 0.31 | 1 |
| Clear mixed forest | 0.90 | 0.13 | 1 |
| Coniferous forest | 26.94 | 3.81 | 1 |
| Dense broad—leaved forest | 163.58 | 23.11 | 1 |
| Dense mixed forest | 27.00 | 3.82 | 1 |
| Field crops in medium to large terraces | 19.14 | 2.70 | 2 |
| Fruit trees | 0.20 | 0.03 | 2 |
| Grassland | 207.04 | 29.26 | 1 |
| Inland marshes | 4.01 | 0.57 | 1 |
| Low density urban tissue | 3.24 | 0.46 | 2 |
| Medium density urban tissue | 1.76 | 0.25 | 2 |
| Mineral extraction site | 0.09 | 0.01 | 2 |
| Non–irrigated field crops | 151.17 | 21.36 | 2 |
| Pond | 79.47 | 11.23 | 0 |
| River | 0.55 | 0.08 | 0 |
| Scrubland | 2.80 | 0.40 | 1 |
| Scrubland with some bigger dispersed trees | 15.36 | 2.17 | 1 |
| Urban expansion site | 0.01 | 0 | 2 |
| Urban sprawl on clear wooded lands | 0.01 | 0 | 2 |
| Urban sprawl on field crops | 1.01 | 0.14 | 2 |
| Urban sprawl on grassland | 1.20 | 0.17 | 2 |

In the case of SWAT, land use and land cover classes were reclassified into SWAT's classes. Particular attention was given to the water (in SWAT terms WATR) classes. As reported by Jalowska and Yuan (2019) [38], the reason for this is that though SWAT allows the creation of HRUs with WATR, water bodies should be modeled either as reservoirs or ponds. Almendinger et al. (2014) and Jalowska and Yuan (2019) [38,48] have shown that an accurate representation of basin processes requires

the integration of the SWAT model's impoundments function. Furthermore, Wang et al. (2008) [77] highlighted the importance of considering impoundments, such as ponds, by testing scenarios of impoundment integration versus impoundment disregarding in a basin covered by only 3% of impoundments. Their results confirmed that simulations were considerably affected even with this small cover. In the same manner, Jalowska and Yuan (2019) [38] simulated different scenarios following integration or absence of impoundments like reservoirs rather than just a normal "water" land occupation class. They noted that disregarding impoundments leads to a series of uncertainties starting by an inaccurate SWAT performance, which in turn leads to inefficient calibration efforts, and overall inaccurate model performance.

After computing the slope, erosivity and soil erodibility indices, the potential soil erosion risk map was constructed. By combining the potential soil erosion risk map to the vegetation cover layer in the "raster calculator" tool and crossing each ones indices, the actual soil erosion risk map was obtained.

## 3. Results

### 3.1. CORINE Erosion Model Outputs for the Claise Basin

In the following sections, the erosion assessment for the Claise is presented and each component is detailed accordingly.

### 3.1.1. Soil Erodibility

Soil erodibility parameters are presented in Figure 3. The dominant textural class in the basin was found to be loam (68.4%) followed by loamy sand (28.75%), while the remainder percentages are clay (2.48%) and sand (0.34%). In terms of texture, since most of the study area is covered by loam with respect to [62], the study area predominantly falls into the highly erodible class. In terms of soil depth, 94.6% of the Claise fits to the slightly erodible class (1000 mm > 750 mm), while the remainder 5.4% rests within the moderately erodible class (300 mm, corresponding to the 250–750 mm class). As far as stoniness is concerned, only 6% of the Claise is under the fully protected cover.

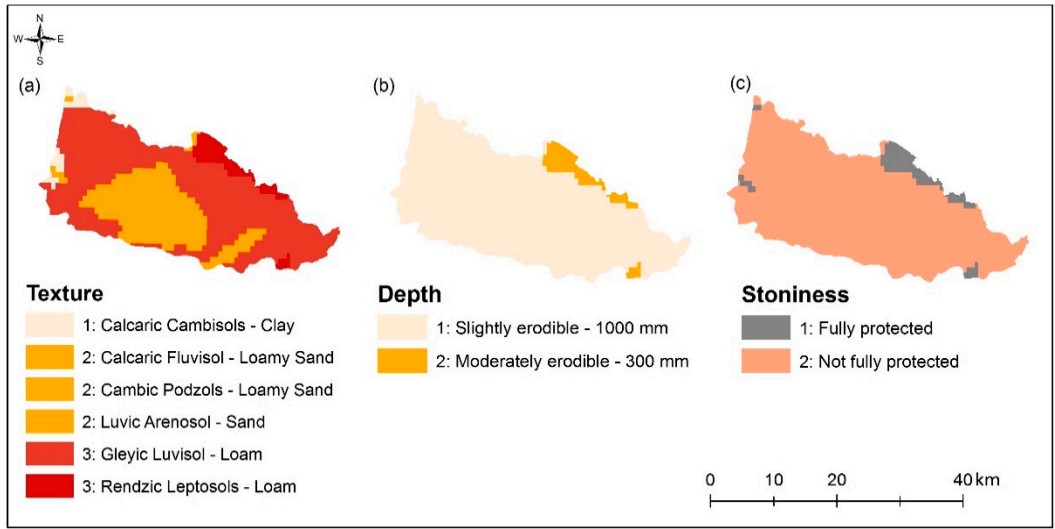

**Figure 3.** Spatial distribution of soil erodibility parameters: (**a**) texture, (**b**) depth, (**c**) stoniness, and class descriptions of the Claise with their respective indices.

Table 4 provides the numerical description of the soil's erodibility components.

**Table 4.** Numerical distribution of soil erodibility parameters and corresponding covered areas.

| Parameter | CORINE Class | Area (km$^2$) | Percentage (%) |
|---|---|---|---|
| **Soil Texture** | 1: Slightly erodible (clay) | 17.53 | 2.48 |
| | 2: Moderately erodible (loamy sand and sand) | 205.88 | 29.12 (28.77 and 0.35) |
| | 3: Highly erodible (loam) | 483.59 | 68.4 |
| | **Total** | **707** | **100** |
| **Soil Depth** | 1: Slightly erodible (>1000 mm) | 668.82 | 94.6 |
| | 2: Moderately erodible (250–750 mm) | 38.18 | 5.4 |
| | **Total** | **707** | **100** |
| **Stoniness** | 1: Fully protected (>10%) | 43.26 | 6.12 |
| | 2: Not fully protected (<10%) | 663.73 | 93.88 |
| | **Total** | **707** | **100** |

By inputting the texture, depth and stoniness parameters into the "raster calculator" tool of ArcGIS, the soil erodibility raster was generated and then reclassified using the "reclassify" tool (Figure 4).

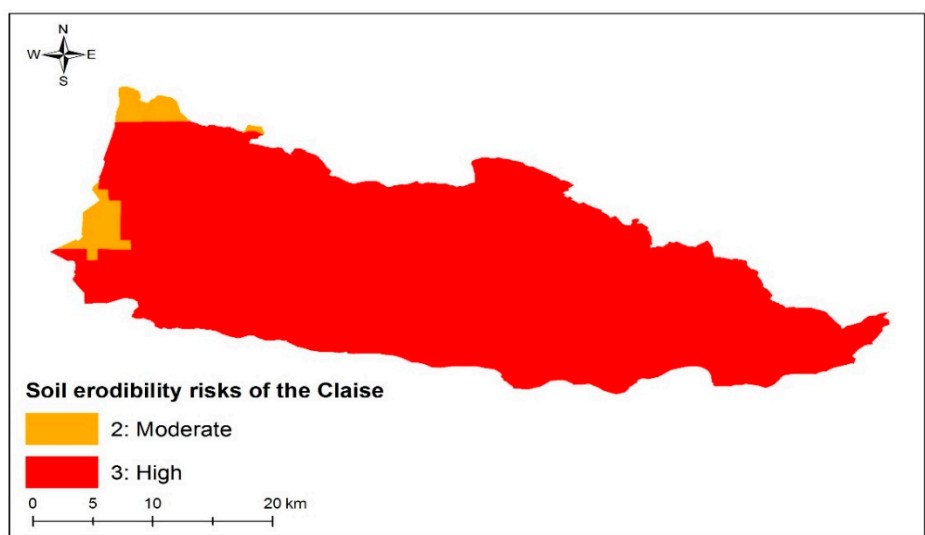

**Figure 4.** Soil erodibility map of the Claise classified according to the CORINE indices.

As seen in Figure 4, the generated soil erodibility map revealed that about 96.83% (684.37 km$^2$) of the study area is covered by highly erodible soils, while the remainder 3.17% (22.63 km$^2$) correspond to moderate-risk areas. This fact is mainly due to the textural distribution of soil and the little stone cover of the Claise basin.

### 3.1.2. Erosivity in a Degraded Oceanic Climate Setting

Meteorological data of the study area is presented in Table 5. As seen, no dry month exists in the Claise and highest temperatures are recorded during the month of July making the basin subject to continuous precipitation.

**Table 5.** Average temperature and precipitation for the Claise basin (1970–2018).

| Month | January | February | March | April | May | June | July | August | September | October | November | December |
|---|---|---|---|---|---|---|---|---|---|---|---|---|
| Average temp. (°C) | 4.7 | 5.2 | 7.8 | 10.4 | 14.3 | 17.9 | 20.1 | 19.7 | 16.1 | 11.8 | 7.7 | 4.8 |
| Precipitation (mm) | 416 | 677 | 1106 | 1526 | 1782 | 2045 | 2108 | 1826 | 1356 | 818 | 482 | 361 |

MFI was found to be 80, corresponding to the low variability class. This signifies evenly distributed rainfall, thus reducing risks of climate-induced soil erosion [78]. The Bagnouls–Gaussen Aridity Index (BGI) on the other hand, was found to be "0", signifying that the study area corresponds to the humid regime as a result of its oceanic influence. In terms of CORINE erosivity, the Modified Fournier Index (MFI) belongs to class 2 variability (low), while BGI corresponds to class 1 (humid). Accordingly, the erosivity index of the Claise watershed was found to be 2, corresponding to the low erosivity class.

### 3.1.3. Topography (Slope): A Reduced Effect in a Flat Setting

Using the "slope" tool in ArcGIS, slope angles were extracted from the DEM. Figure 5 displays the slope of the Claise basin and the adapted reclassified raster following CORINE classification.

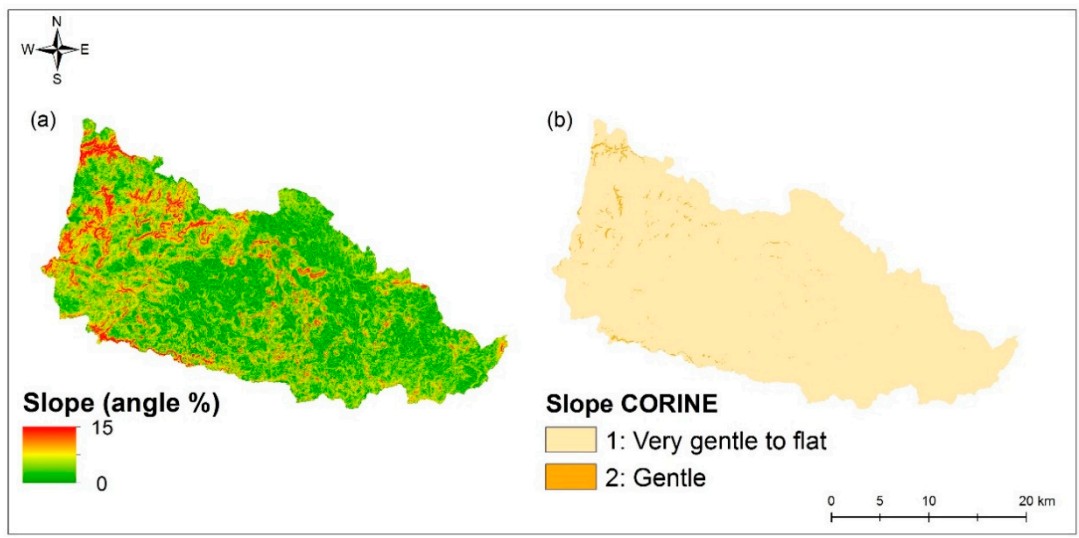

**Figure 5.** (**a**) Slope angle of the Claise and (**b**) the corresponding CORINE indices description.

From Figure 5, it can be observed that the Claise presents a dominantly flat topography with a maximum slope of 15°. In CORINE terms, 99.3% (702.051 km$^2$) of the study area corresponds to "very gentle to flat" slope class (<5°) and 0.7% (4.94 km$^2$) to the "gentle" slope category (5°–15°). The particular setting of low relief and small slope significantly plays a role in the reduction of erosion generated by runoff [79], despite the high soil erodibility risks.

### 3.1.4. Potential Soil Erosion Map

By overlaying soil erodibility (Figure 4), topography and erosivity, using the "raster calculator" tool, the potential soil erosion risk map was obtained (Figure 6), following the formula presented in Figure 2.

From Figure 6, low-erosion risks present 3% (21.22 km$^2$), moderate risks present 96.5% (682.25 km$^2$) and high risks form 0.5% (3.53 km$^2$). These percentages, however, reflect only the potential soil erosion risks which according to CORINE (1992) [62] do not take into consideration the vegetative cover at this stage. The slope effect and equal repartition of precipitation, as reflected by the MFI values and the absence of arid periods (low BGI), are potentially responsible for moderating the soil erodibility map yielding the dominantly moderate potential soil erosion risks.

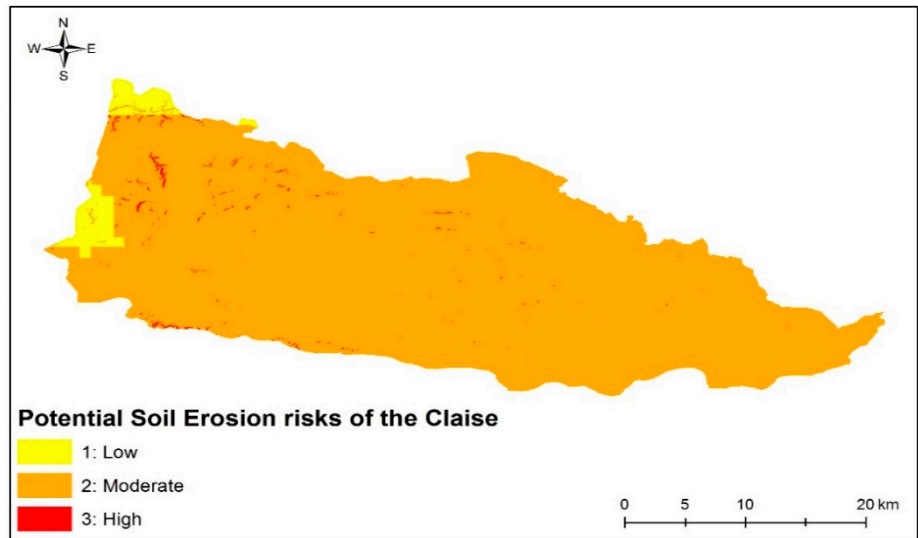

**Figure 6.** Potential soil erosion (PSE) risk map of the Claise basin.

### 3.1.5. Vegetation Cover: The Presence/Absence Effect of Ponds

This factor is the focal point of this study since by slightly changing this parameter the outcome changes significantly. At this point, two vegetative cover scenarios are presented: the first presents the actual setting of the Claise accounting for the presence of ponds, while the second simulates a scenario where ponds are removed to assess the difference in erosion outcomes with and without their presence. This last step allowed to quantify the impact of ponds on erosion in the Claise basin. In the second scenario, the land occupation group "ponds" was changed to their surrounding class (grasslands).

The large area of ponds, which makes up around 11% of the Claise (Table 3), displays their potential role as modifiers basin processes. Figure 7 presents the two considered vegetation covers as inputs for crossing with the potential soil erosion risk map to yield two different actual soil erosion risk maps. These maps were then compared to evaluate the effect of the presence and absence of ponds and their role on erosion.

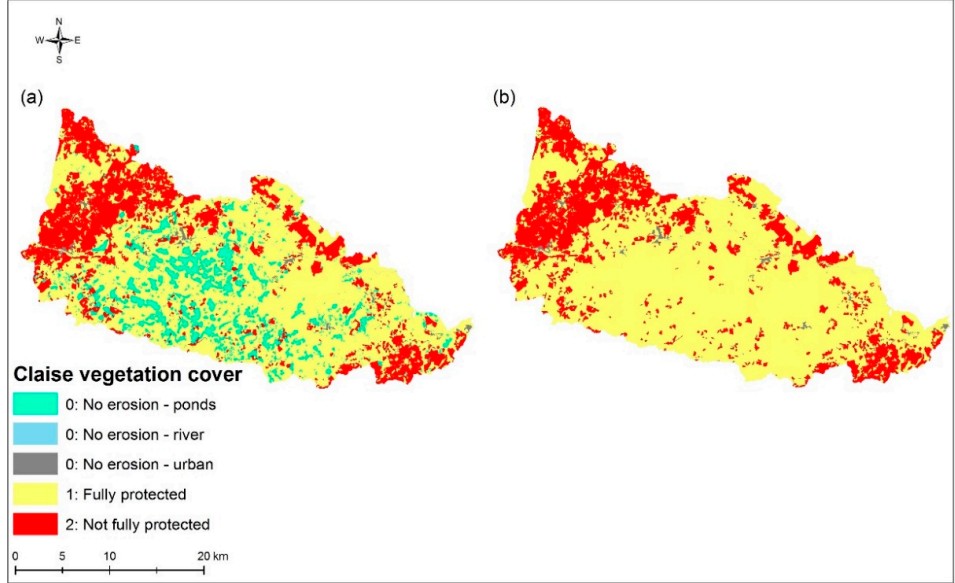

**Figure 7.** (**a**) Current Claise vegetation cover and (**b**) alternative vegetation cover and corresponding CORINE classification; 1: fully protected, 2: not fully protected.

Class 0 refers to land cover categories that are not considered in CORINE; these categories are urban areas and water bodies, while classes 1 and 2 refer to the fully protected and not fully protected covers. From Figure 7b, 72.5% of the Claise basin corresponds to the fully protected class, while 27.5% of the study area is occupied by not fully protected cover. To assess the impact of pond presence, the second scenario of replacing ponds by their surrounding dominant cover was performed.

### 3.1.6. Actual Soil Erosion Maps Under Current and Alternative Scenarios

The two actual soil erosion risks maps were produced by multiplying the respective indices of the potential soil erosion risk map and the two vegetation cover scenarios using the "raster calculator" tool. Figure 8 reveals the outcome under both scenarios.

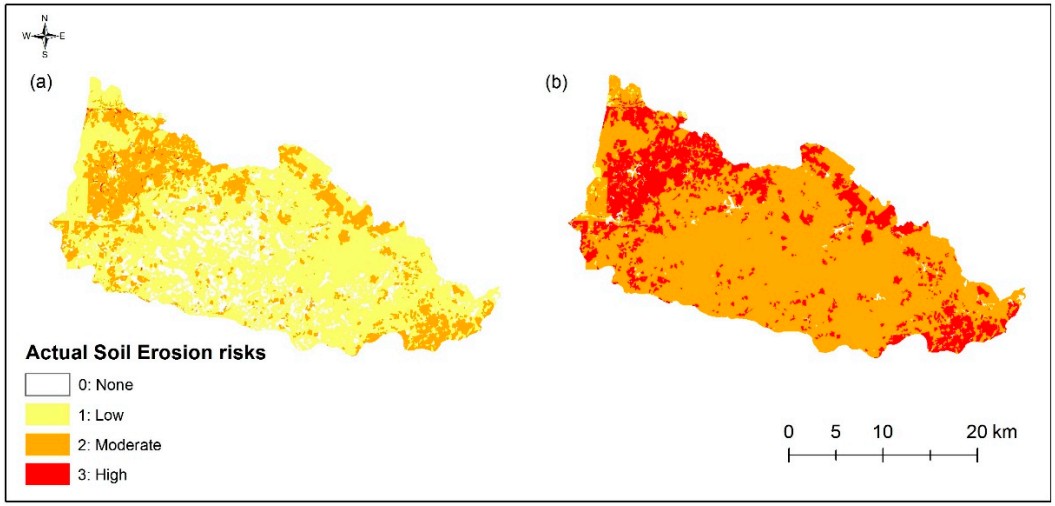

**Figure 8.** Actual soil erosion risk map of the Claise under (**a**) current vegetation cover and (**b**) the alternative pondless scenario.

The Actual soil erosion map was cross-checked against: Institut National de la Recherche Agronomique (INRA) 2000 erosion maps and the combined GIS sol–INRA–SOeS 2011 maps [80]. These were produced following the Modèle d'Évaluation Spatiale de l'Aléa d'Érosion des Sols (MESALES). The comparison between the INRA maps, and the produced actual soil erosion risk map, is presented in Table 6. By this comparison, it is concluded that there is a good agreement between these maps. In addition, since INRA maps are not completely adequate to be considered at the basin scale [80], the established erosion maps are considered for the no-erosion zones, overcoming, this way, the challenge of coarse representation.

**Table 6.** Verification of the established erosion map.

| Classes | Low | Moderate | High | Validation Points |
|---------|-----|----------|------|-------------------|
| Low | 76 | 10 | 0 | 86 |
| Moderate | 15 | 14 | 8 | 37 |
| High | 0 | 0 | 11 | 11 |
| Total | 91 | 24 | 19 | 134 |

As can be seen from Table 6, a total of 134 validation points were chosen. These were divided into 86, 37, and 11 low-erosion, moderate-erosion, and high-erosion zones validation points, respectively. A large part of the moderate-erosion class was misinterpreted as the low-erosion class. This discrepancy is due to the fact that the no-erosion zones do not exist in the INRA maps, but are instead classified as low-erosion zones. Therefore, the error margin in the moderate-erosion class from Table 6 is justified by the finer scale representation of the produced maps, compared to the INRA maps. The overall

accuracy was determined to be 75%, while the computed Cohen's kappa coefficient [81] was found to be 0.7; this indicates a substantial agreement between the INRA maps, and the produced actual soil erosion risk map. The kappa coefficient was used since it tests inter-rater reliability; i.e., the coefficient represents the extent to which the generated data are correct representations of the measured data. In the case of this study, the generated data is the actual soil erosion risk map, while the measured data consists of the validation points obtained from the INRA maps.

From Figure 8, three main results can be drawn:

(1) The role of vegetation cover in changing erosion risks is solidified. This is particularly reflected by the setting of the Claise basin due to the agricultural and grass cover. By comparing the actual soil erosion risk map with the potential soil erosion risk map and statistics, a shift of erosion risk classes is observed. As mentioned in Section 3.1.4, considering the potential soil erosion risks, and ignoring the vegetation cover, low, moderate and high-risk areas take over 3%, 96.5% and 0.5% of the total basin area, accordingly. When the vegetation cover layer was taken into account the resulting actual soil erosion risk shifted to 65.66%, 21.68% and 0.18%, for low, moderate and high-risk areas. These observations solidify that vegetative cover is the most influential aspect for erosion assessment. In further detail regarding the vegetation cover layer, areas corresponding to agricultural classes are seen to have higher erosion risks than areas with different land cover types. This is in agreement with Verheijen et al. (2009) [82] observations that despite the considerable effect of soil type, topography and climatic conditions, the major influencer of soil erosion is the vegetative cover, especially cultivated areas.

(2) The remainder 12.48% of the actual soil erosion risk map is the no-erosion zone. As seen in Figure 7a, most of the no-erosion zone corresponds to the concentration area of ponds, while the remainder 1.48% represents the Claise River. The ponded area represents 88.23 km$^2$ of the Claise under no risk of erosion, making these ponds a counter-erosion zone.

(3) At a graphical scale, a complete shift from low to moderate risks, in the greatest part of the basin, is observed. Table 7 presents the statistical difference between the actual soil erosion risks with current vegetation cover and those of the pondless scenario. From Table 7, the effective role of ponds as an erosion counter-measure is revealed.

**Table 7.** Actual soil erosion risk (ASE) for the Claise basin under current and simulated vegetation cover.

| Erosion Risks | ASE with Current Vegetation Cover—Area (km$^2$) | ASE with Simulated Vegetation Cover (Absence of Ponds)—Area (km$^2$) | ASE with Current Vegetation Cover—Percentage (%) | ASE with Simulated Vegetation Cover (Absence of Ponds)—Percentage (%) |
|---|---|---|---|---|
| None | 88.23 | 7.92 | 12.48 | 1.12 |
| Low | 464.21 | 3.6 | 65.66 | 0.52 |
| Moderate | 153.27 | 543 | 21.68 | 76.8 |
| High | 1.272 | 152.4 | 0.18 | 21.56 |

Additionally, the impact of ponds on erosion at the scale of the basin is revealed. Not only did the no-erosion and low-erosion classes decrease by 11.36% and 65.14%, in the absence of ponds scenario, but also the moderate and high-erosion risks increased by 55.12% and 21.38%, respectively. These changes are due to several reasons:

(1) The most evident reason is that ponds effectively and directly nullify splash erosion in the areas they occupy.

(2) Their widespread, yet dense, positioning throughout the basin, counteracts runoff erosion in a twofold way: first, by intercepting eroded soils by overland flow, retaining this way, the transported material and preventing them from reaching the streams, and second, by slowing surface runoff and thus, abating its erosive force. Despite the fact that the low-slope topography does not particularly favor runoff erosion, let alone high velocities of overland flow, this obviously has some effect, especially in cases of intense rainfall events.

(3) Their dense aggregation in the basin attributes them the role of cascade check dams, containing sediments.

(4) Their large density and chain sequence where the retention effect is greatly amplified (factor of 2179 ponds) [38].

(5) The highly erodible setting of the basin resulting from a challenging pedology.

### 3.2. Sediment Transport in a Limnologically Rich Setting

Sediment transport in the Claise was simulated using the SWAT model. The model was calibrated using the SUFI-2 algorithm of SWATCUP following Jalowska and Yuan (2019) [38] proposed sequential calibration for settings characterized by the presence of water impoundments. As mentioned previously, due to the unavailability of sediment measurements for this study, only a hydrologic calibration was performed with the most related sensitive parameters (Table 8). Results of the hydrologic calibration yielded an $R^2$ of 0.7 during calibration and 0.67 during validation.

**Table 8.** Sensitivity analysis for calibration of the SWAT model.

| Parameter Name | $t$-Stat | $p$-Value | Fitted Value |
|---|---|---|---|
| R__WET_NVOL.pnd | −37.52 | 0.00 | 0.823 |
| V__GW_DELAY.gw | −31.39 | 0.00 | 52.325 |
| R__PND_SED.pnd | 1.62 | 0.11 | 318.100 |
| V__GWQMN.gw | 1.59 | 0.11 | 1.461 |
| R__SPCON.bsn | −1.35 | 0.18 | 0.005 |
| R__CN2.mgt | 1.29 | 0.20 | 0.564 |
| R__PND_NSED.pnd | 1.23 | 0.22 | 2420.81 |
| R__USLE_K.sol | −0.97 | 0.33 | 0.209 |
| V__ALPHA_BF.gw | −0.90 | 0.37 | 0.529 |
| R__NDTARG.pnd | 0.44 | 0.66 | 49.890 |
| R__PND_FR.pnd | 0.15 | 0.88 | 0.644 |
| R__USLE_P.mgt | −0.11 | 0.91 | 0.188 |
| R__IGRO.mgt | −0.04 | 0.97 | 0.080 |

As a result of an exhaustive hydrologic calibration, the average sediment yield of the Claise basin for the studied period under current and alternative conditions is presented in Figure 9.

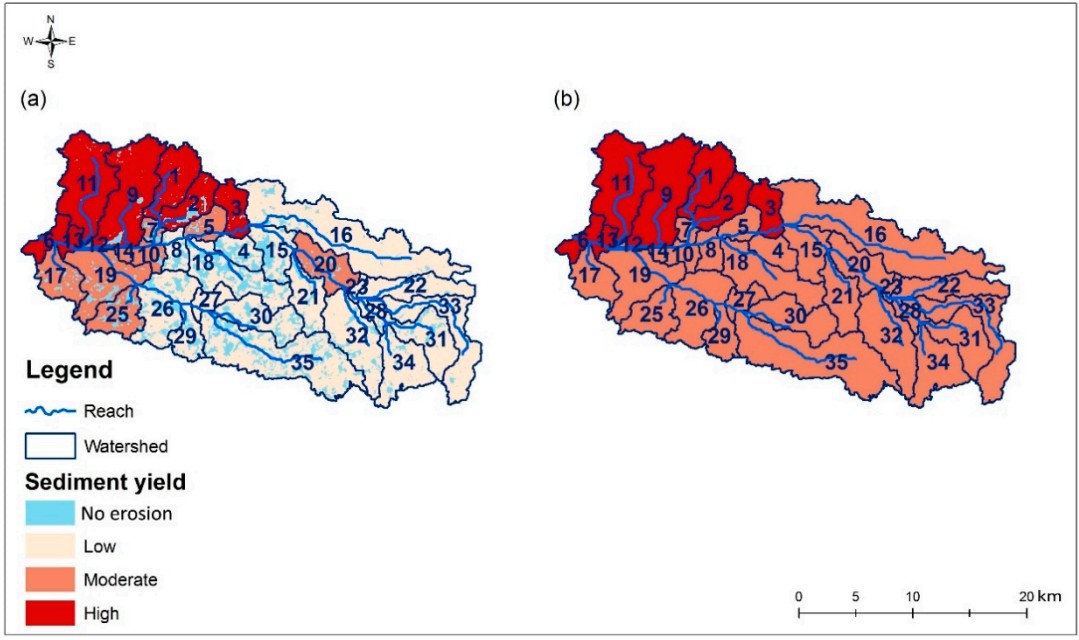

**Figure 9.** Sediment yields of the Claise basin under (**a**) current conditions and (**b**) the pondless scenario.

As can be seen from Figure 9, low sediment yields correspond to ponded sub–basins in contrast to pondless sub-basins which present lower soil loss rates. However, this gradient varies according to the setting of the ponds in these sub–basins. Despite that sub-basins 1 and 9 contain a small number of ponds, these correspond to areas of high sediment yield as a result of their dominant agricultural cover and their location in the basin's steepest areas. This shows that the presence of ponds alone is not sufficient to alter the sediment yields. In fact, the setting of ponds under the form of a collective dense network is the main modifier of sediment transport patterns. In order to further highlight the impact of ponds, alternative scenario testing was performed where a pondless land occupation setting was re-inputted to the calibrated SWAT model in order to determine subsequent sediment transport changes. Results of sediment yields, highlighting the impact of ponds, are presented in Table 9.

**Table 9.** Shifts of sediment yields as a result of pond presence or absence.

| Sediment Yield | Pond Presence (Area km$^2$; Percentage %) | Pond Absence (Area km$^2$; Percentage %) |
|---|---|---|
| Low | 365.07; 63.31 | 0; 0 |
| Moderate | 89.23; 15.47 | 454.30; 78.79 |
| High | 122.27; 21.20 | 122.27; 21.20 |

In the low sediment yield sub-basins, where the vast majority of ponds lies, occupying a large portion of the sub-basins they are located in, the land cover change from pond to grasslands is far too great and the effect of ponds presence/absence is quite obvious. Accordingly, the role of ponds as integral features of the landscape processes is solidified. Having determined their effect, an increased exposure for their integration into management plans is recommended, while an understanding of basin scale sediment dynamics offers insights regarding the role of ponds in water resources. Such implications could matter not only for the Claise and similar basins, but also for expectations regarding the landscape role of sediment retention basins, and for the general understanding of sediment transport in rain-dominated contexts. This, in turn, could be considered as a contribution to hydrologic modelling efforts, where small waterbodies, especially artificial ones, are often neglected, or even smoothed out, from digital elevation models.

## 4. Conclusions

An investigation of the pond-induced effects on soil erosion and sediment transport of limnologically rich basins was presented. Through this task, recommendations of the European framework for the Thematic Strategy on Soil Protection were addressed by revealing the different levels of soil loss, represented by providing an insight to the investigation of erosion-prone regions and sediment yield zones of different levels. Furthermore, recommendations of the DCE regarding the behavioral understanding of hydromorphological alternating factors (ponds as hydro-sedimentary elements), at the basin scale, were also considered. Despite the Claises' weakly structured pedology, resulting in high erodibility, the Claise was found to have low erosion risks and sediment transport rates due to several reasons, like the evenly distributed rainfall, the relatively flat topography, and most importantly due to the presence of its dense pond network that acts as a natural measure against soil erosion. This was solidified by simulating a scenario where ponds were substituted by grasslands. After replacing ponds by a protective vegetative cover, all erosion risks and sediment yield classes of the Claise significantly varied: no and low-erosion risk zones decreased, while moderate and high-erosion risk zones increased. Regarding sediment transport, the replacement of ponds by grasslands led to a complete disappearance of low sediment yield zones and considerable increases of moderate and high sediment yield zones. Accordingly, the "safe" soil loss status of the Claise can be attributed to the low soil loss rates of the Claise basin to the presence of these ponds. Despite their protection against soil erosion, however, their presence in very large numbers might cause a distortion in the sediment balance of the underlying rivers. Such cases may lead to sediment starvation and force the river to

engage in increasing streamed and bank erosion [83]. Therefore, the use of the presented approach may serve as an efficient tool towards the orientation of future decisions regarding the proliferation or cease of ponds depending on their effect.

**Author Contributions:** Conceptualization: R.N. and M.J.A.S.; data curation: M.J.A.S.; methodology: M.J.A.S. and R.N.; project administration: R.N., C.A., and M.K.; supervision: R.N., K.K., C.A., and M.K.; validation: M.J.A.S.; visualization: M.J.A.S. and K.K.; writing—original draft: M.J.A.S. and K.K.; writing—review and editing: M.J.A.S., K.K., R.N., C.A., and M.K.

**Funding:** This research is part of a PhD thesis funded by the National Council of Scientific Research—Lebanon (CNRS—L), Agence Universitaire de la Francophonie (AUF), Lebanon, and the Lebanese University. It is also part of the Dynétangs project funded by the French Centre–Val–de–Loire region.

**Acknowledgments:** The authors would like to express their gratitude for the funding agencies, the editor, and reviewers for leveraging the quality of this work and to the Brenne Natural Park for their help.

**Conflicts of Interest:** The authors declare no conflict of interest.

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
