# Peer review of "Assessing the Impact of Man–Made Ponds on Soil Erosion and Sediment Transport in Limnological Basins"

_water, doi:10.3390/w11122526_

Round 1

Reviewer 1 Report

The manuscripts present the results of overland erosion and the effect of pons on soil erosion by utilizing two models.

The project background, the models' descriptions, and the input data are well presented. I have the following specific comments on the results:

Line 188: the authors claim that even in the absence of sediment data, hydrologic calibration of the model(s) would be sufficient. The driving forces for overland erosion and sediment transport are indeed runoff and streamflow, but if not sediment source is available, erosion will not occur. So, the existence of erosive force does not necessarily result in sediment yield if the surface is not erodible (e.g., covered with dense vegetation or rock material). A model that is not fully calibrated may not be suitable as a  tool for decision-makers. Line 343-345: It seems that the main objective of the paper was to assess the role of ponds in erosion? Why in this section is the effect of vegetation in erosion ignored? It is also very confusing for the reader to compare fig 4 and fig 6, with almost exactly the same titles and captions, and both presenting" soil edibility risks of the Claise" without clarifying the difference between them. Line 348-403: It is well-document in the literature that the vegetation plays a significant role in erosion control, so why the watershed with and without vegetation cover is simulated. It could be more reasonable if this work instead presented the relation between types and density of vegetation on erosion rate within the watershed. Line 412: Referring to ponds as "counter-erosion zone" might be an accurate terminology because the shoreline of ponds is susceptible to erosion and, therefore, could be considered as a sediment source. However, ponds are obviously sediment sinks that abate watershed sediment yield. Table 7: Comparing ASE for with and without ponds for low erosion risks are reported 464.21 and 3.6 km2. Since the ponds are only 79.47 km2, why is there a significant decrease in areas with Low-risk ad significant increase in area with Moderate risk? One may expect to see a decrease in the area with None (because of the absence of ponds) and an increase in area with Low risk (due to replacing pond areas with grassland). The reasons listed in Lines 424-435 are very generic and may not apply to this study. For example, it does not seem that the CORINE model solves any sediment budget to account for sediment trap in ponds as referred to in item (2). Also, referring to ponds as run-off-the river-dams counter is a counter-argument for sediment intercept by dams. A run-off-the river-dam does not trap sediment, so it has minimal to no effect of abating sediment yield. The results from the SWAT model are presented too briefly comparing to that for the CORINE model. Table 9: Unlike in Table 7, the None-erosion area is not presented in this paper. So the results from these two models can not be directly compared.

Author Response

The Authors would like to thank the Editor and the Reviewers for carefully reading the manuscript and for their constructive comments. The Authors deeply appreciate the recommendations of the Reviewers and acknowledge their importance for improving the paper. Accordingly, the Authors have revised the manuscript carefully, clarified all controversial issues, highlighted references, corrected errors and answered all the related questions. All the performed modifications and corrections are explained below and were incorporated in the modified text.

Reviewers’ comments are listed in bold, the Authors’ replies are in normal font and modified changes are in italic  

Reviewer 1

1) Line 188: the Authors claim that even in the absence of sediment data, hydrologic calibration of the model(s) would be sufficient. The driving forces for overland erosion and sediment transport are indeed runoff and streamflow, but if not sediment source is available, erosion will not occur. So, the existence of erosive force does not necessarily result in sediment yield if the surface is not erodible (e.g., covered with dense vegetation or rock material). A model that is not fully calibrated may not be suitable as a tool for decision–makers.

Thank you for pointing out this issue. The Authors agree with the comment and respond as follows:

a) The Authors do not claim that once runoff is generated erosion will occur. This reference is made solely with regard to the modeling part of the study. In that sense, the model has all the necessary information (meteorological data, soil data, land cover data, topography) to determine whether or not erosion will occur. Regarding the erodibility of the surface, a detailed analysis of the soil erodibility component of the CORINE model was presented. Accordingly, the soil’s erodibility was assessed using soil texture, depth and stoniness (rock material) [please refer to Figure 2]. These factors are presented in Figure 3 and in Table 4 where the soil’s erodibility is numerically presented. In addition to these factors, the soil input was also used in the SWAT model. Hadn’t the surface been erodible (which in SWAT is assessed using the MUSLE equation) the model would have given null values for sediment transport. In lines 150–153, the components of the MUSLE equation were explained, where the factors listed in this comment (vegetation and rock material) are accounted for in the K, F, and C factors of the MUSLE equation. b) Indeed a full calibration (hydrologic + sedimentary) would provide a completely reliable model. However, sediment data is a limitation in most related studies, as soil erosion measurements, either in the form of gross erosion or net erosion or in the form of total sediment discharge in stream segments, is a very difficult and laborious task. Even in cases where there is such data, it is usually in the form of sparse discrete measurements and rarely in the form of continuous sediment graphs, where they could be used for a robust and meaningful calibration.

For this reason, the Authors referred to the SWAT model documentation guide Arnold et al. (2012) [line 196] and to Hallouz et al. (2018) [line 198], to highlight the potential of a well calibrated hydrologic model, providing a reliable degree of accuracy to the sedimentary output. Further, the Authors followed the calibration sequence of Jalowska and Yuan (2019) which describes sequentially the steps to perform in a basin loaded with impoundments (here ponds). This was done for two main reasons:

The ponds exceed 2000, in number. Therefore, measurements of sediment transport for each pond are not possible. By incorporating all elements of Jalowska and Yuan’s (2019) calibration sequence, a complete representation of the pond physical processes is ensured, as explained in lines 95–98. This, in turn, results in a reliable degree of accuracy for representing the investigated pond processes.

Considering all this and with response to this comment, the Authors presented the sediment yield map in a qualitative form rather than quantitatively, to avoid any unreliable information. The Authors chose to display the sedimentary output of the Claise SWAT model under the form of a map with different sediment yield levels (low, moderate and high), since the hydrologic calibration is adequate enough to present the different sediment yield levels and to avoid imprecise dissemination of information. Nonetheless, it is true that this poses some short limitations. Decision–makers, as well as private or public entities that might consult the present study, in the future, should take this limitation into consideration. In any case, the reasoning of the Reviewer is meaningful.

Accordingly a short text was added in lines 199–207 as follows: However, the availability of sediment data is often a constraint in relative studies, as measurements, either in the form of gross or net erosion or in the form of total sediment discharge, in streams, are often nonexistent. Even in cases where such data is available, it is in the form of sparse discrete measurements and rarely in the form of continuous sediment graphs, where they could be used for a robust calibration. In the case of the Claise, this would be a very challenging task, given the large number of ponds. For this reason, SWAT was calibrated according to the calibration scheme of Jalowska and Yuan (2019) [39], where a complete representation of an impoundment effect (here ponds) is ensured. Nonetheless, the absence of sediment records poses some short of limitation that needs to be considered by the decisionmakers of the basin.” 

2) Line 343–345: It seems that the main objective of the paper was to assess the role of ponds in erosion? Why in this section is the effect of vegetation in erosion ignored? It is also very confusing for the reader to compare fig 4 and fig 6, with almost exactly the same titles and captions, and both presenting" soil edibility risks of the Claise" without clarifying the difference between them.

a) With respect to this comment, the following modification was made in lines 359–361: “These percentages, however, reflect only the potential soil erosion risks which according to CORINE (1992) [62] do not take into consideration the vegetative cover at this stage. Moreover, the role of the vegetation cover – under current and alternative scenarios – is considered in the Actual Soil Erosion Risk Map, as indicated in Figure 2 and explained in lines 389–391 (previously 372–374). b) Figure 4 presents the soil erodibility of the Claise, meaning how prone to erosion are the soils of the study area. This raster groups’ soil texture, depth and stoniness as indicated in Figure 2 (i.e: the physical properties of the soil alone). Figure 6, on the other hand, depicts the Potential Soil Erosion Risks of the basin which is a product of soil erodibility, climate erosivity, and slope, as also indicated in Figure 2. In lines 313–315 (previously 299–301), we explained that figure 4 was obtained by inputting soil texture, depth and stoniness into the raster calculator tool, while for Figure 6 we explained that soil erodibility, topography and erosivity were overlain in the raster calculator tool [lines 351–353 (previously 337–338) ] and made a reference to Figure 2. However, in order to clear any ambiguities, a modification, in line 351, was made: “By overlaying soil erodibility (Figure 4), topography and erosivity, using the “raster calculator” tool, the Potential Soil Erosion risk map was obtained (Figure 6), following the formula presented in Figure 2.”

3) Line 348–403: It is well–document in the literature that the vegetation plays a significant role in erosion control, so why the watershed with and without vegetation cover is simulated. It could be more reasonable if this work instead presented the relation between types and density of vegetation on erosion rate within the watershed.

a) In this paper simulations were performed using two vegetation cover scenarios: The first is the land occupation pattern of the study area as presented in Table 3, while the other consists of a simple replacement of ponds by grasslands. Accordingly, this was highlighted in several lines of the manuscript, e.g: lines 20–21, line 26, lines 101–103, lines 365–369.

It can also be observed from Figure 7 that the land occupation was overall kept the same except for the removal of ponds. As explained through different parts of the manuscript, the aim was to determine the impact of ponds by assessing their effect (presence) and comparing it with changes issued by their absence, i.e. the pondless scenario. Evaluating the presence/absence effect provides decision makers a “what to expect” insight in the case of any modification.

Thank you for this insight; this is exactly the idea that we are trying to develop in another manuscript.

4) Line 412: Referring to ponds as "counter–erosion zone" might be an accurate terminology because the shoreline of ponds is susceptible to erosion and, therefore, could be considered as a sediment source. However, ponds are obviously sediment sinks that abate watershed sediment yield.

Ponds – as micrography of lakes – have a similar sedimentary behavior with lakes, mainly in the sense that they constitute sediment sinks. As every unprotected part of soil surface, the shoreline of ponds could be susceptible to erosion. However, the detached soil from the ponds’ shoreline directly sinks into the ponds and does not contribute to the soil mass that is transported downslope by surface runoff. 

In addition, the edges of the ponds in the Claise are poorly developed, and hence are not dynamic. Therefore, they cannot play a real role in the soil/sedimentary dynamics of the study area. Further, since the ponds are located in the flat areas of the basin, even if they witness bank erosion, the eroded particles directly sink into the pond where they are immediately retained.

Moreover, as mentioned in the study area description, the ponds are grouped in the form of an interconnected network, therefore, the individual retention effect is amplified by a factor of 2000. Accordingly the ponds of the Claise are points of sediment load retention.

5) Table 7: Comparing ASE for with and without ponds for low erosion risks are reported 464.21 and 3.6 km2. Since the ponds are only 79.47 km2, why is there a significant decrease in areas with Low–risk and significant increase in area with Moderate risk? One may expect to see a decrease in the area with None (because of the absence of ponds) and an increase in area with Low risk (due to replacing pond areas with grassland).

This discrepancy may be one of the drawbacks of the CORINE model, where the model is only flexible for the land occupation input. Consequently, a replacement of ponds by grasslands was performed without a replacement of the underlying soil types. As discussed previously, the basin’s soils are highly erodible. Nonetheless, since ASE maps are obtained by crossing the indices of the vegetation cover with those of the PSE, the difference was due to the following reason:

Ponds in the vegetation cover have an index of 0, the pond dense zone in the PSE map has an index of 2 (moderate) the multiplication of the pond index and the PSE index resulted in 0 meaning a no erosion zone. When the pond class was replaced by grasslands, the index of vegetation cover shifted to 1, meaning that the index of the new cover x the PSE index (2) yielded an index of 2, meaning a moderate zone.

Had the CORINE model been used alone, this would have significantly raised questions regarding the results. However, the simulations of the calibrated SWAT model, when the pond cover was removed, also shifted to moderate yield zones. A similar shift between the CORINE and the calibrated SWAT model grants the outputs of both models certain degrees of reliability.

6) The reasons listed in Lines 424–435 are very generic and may not apply to this study. For example, it does not seem that the CORINE model solves any sediment budget to account for sediment trap in ponds as referred to in item (2). Also, referring to ponds as run–off–the river–dams counter is a counter–argument for sediment intercept by dams. A run–off–the river–dam does not trap sediment, so it has minimal to no effect of abating sediment yield. The results from the SWAT model are presented too briefly comparing to that for the CORINE model.

a) Regarding the comment for the CORINE model, the Authors agree and accordingly modified item 2 into: “Their widespread, yet dense, positioning throughout the basin, counteracts runoff erosion in a twofold way: first, by intercepting eroded soils by overland flow, retaining, this way, the transported material and preventing them from reaching the streams, and second, by slowing surface runoff and thus, abating its erosive force.” Lines 451–454. b) The Authors agree with the comment about the run–off–the–river dams effect and rephrased into: “Their dense aggregation in the basin attributes them the role of cascade check dams, containing sediments”. (line 457–458) c) In response to this comment, the discussion of the SWAT model results has been further developed as follows (lines 493–501): “Accordingly, the role of ponds as integral features of the landscape processes is solidified. Having determined their effect, an increased exposure for their integration into management plans is recommended, while an understanding of basin scale sediment dynamics offers insights regarding the role of ponds in water resources. Such implications could matter not only for the Claise and similar basins, but also for expectations regarding the landscape role of sediment retention basins, and for the general understanding of sediment transport in rain–dominated contexts. This, in turn, could be considered as a contribution to hydrologic modelling efforts, where small waterbodies, especially artificial ones, are often neglected or even smoothed out from digital elevation models.”

7) Table 9: Unlike in Table 7, the None–erosion area is not presented in this paper. So the results from these two models cannot be directly compared

Thank you for highlighting this. The Authors agree that no direct comparison can be made between both models. For this reason, the outcomes were separated into two different tables (Table 7 and Table 9). The no–erosion zone corresponds to pond zones of the basin. While the CORINE model simulates erosion risks for the whole basin, SWAT simulates the sedimentary behavior of sub–basins that are defined during the watershed delineation. As explained previously, MUSLE was used for the totality of the basin, while the sedimentary behavior of ponds was accounted for by SWAT using Equation 2 (line 157). However, the Authors completely agree that no direct comparison can be made between both models. In order to address the comment, an additional class: “no erosion” was added in the legend of Figure 9.

Reviewer 2 Report

The manuscript titled “Assessing the impact of man-made ponds on soil erosion and sediment transport in limnological basins” presents a study conducted in the French Claise basin using CORINE and SWAT models. The simulations were evaluated with both erosion map and basin discharges, showing reasonably well agreement. The results highlight the importance of ponds in modifying soil erosion and sediment transport through the comparison of two scenarios. Generally, the study is well designed, and the manuscript is well written. I therefore recommendation publication after incorporating the following comments.

Line 26: “under the alternative scenarios”: it is unclear what the alternative scenario is in the abstract. Please clarify it. Lines 65–67: “… results showed that ponds contribute significantly to several basin related processes. Examples … and climate”: Please point out that emission of greenhouse gases is also one of the important processes in the ponds. See Xiao, et al. (2014). Gas transfer velocities of methane and carbon dioxide in a subtropical shallow pond. Tellus B: Chemical and Physical Meteorology, 66(1), 23795. https://doi.org/10.3402/tellusb.v66.23795 Table 1: The simulation was conducted in 1970–2018, but the soil map and LULC map in 2008 and 2018 were used. How will such temporal inconsistency influence the accuracy of the results? Line 264: “[76, 38] have …”: Such in-text citation is inappropriate. Please revise. Line 283: Figure 3 is not “soil erodibility”. Should be soil erodibility parameters. Line 370: “replacing ponds by their surrounding cover”: by the dominant cover? Table 6: According to the results, a large part of the moderate class is misinterpreted as low class. Some explanation should be provided here. Line 413: “a complete shift from low to moderate risks is observed …” in?

Author Response

Manuscript title: Assessing the impact of man–made ponds on soil erosion and sediment transport in limnological basins

Manuscript ID: water–632156

Manuscript Authors: Mario J. Al Sayah, Rachid Nedjai, Konstantinos Kaffas, Chadi Abdallah* and Michel Khouri

 The Authors would like to thank the Editor and the Reviewers for carefully reading the manuscript and for their constructive comments. The Authors deeply appreciate the recommendations of the Reviewers and acknowledge their importance for improving the paper. Accordingly, the Authors have revised the manuscript carefully, clarified all controversial issues, highlighted references, corrected errors and answered all the related questions. All the performed modifications and corrections are explained below and were incorporated in the modified text.

Reviewers’ comments are listed in bold, the Authors’ replies are in normal font and modified changes are in italic  

Reviewer 2

1) Line 26: “under the alternative scenarios”: it is unclear what the alternative scenario is in the abstract. Please clarify it.

Thank you for highlighting this. The following modification was made:

Line 26: “Under the alternative pondless scenario, erosion risks shifted to …”

2) Lines 65–67: “… results showed that ponds contribute significantly to several basin related processes. Examples … and climate”: Please point out that emission of greenhouse gases is also one of the important processes in the ponds. See Xiao, et al. (2014). Gas transfer velocities of methane and carbon dioxide in a subtropical shallow pond. Tellus B: Chemical and Physical Meteorology, 66(1), 23795. https://doi.org/10.3402/tellusb.v66.23795   

This contribution was added in lines 65–68: “Examples of these processes are sediment interception [32], removal of pollutants for river protection [33], nutrient recycling [34], greenhouse gases emission [35], regulation of hydrological flows [36], biogeochemistry [31] and climate [37].”

b) Reference numbers and the reference list were updated accordingly. c) The citation for the article can be found in lines 594–569 of the reference list as follows: Shangbin X.; Hong Y.; Defu L.; Cheng Z.; Dan L.; Yuchun W.; Feng P.; Yingchen L.; Chenghao W.; Xianglong L.; Gaochang W.; Li L. Gas transfer velocities of methane and carbon dioxide in a subtropical shallow pond, Tellus B: Chemical and Physical Meteorology 2014; 66(1), 23795. DOI: 10.3402/tellusb.v66.23795

3) Table 1: The simulation was conducted in 1970–2018, but the soil map and LULC map in 2008 and 2018 were used. How will such temporal inconsistency influence the accuracy of the results?

Thank you for this very important comment. Indeed, the simulations were performed for the years 1970–2018, while the soil and land occupation databases correspond to the years 2008 and 2018. However, the simulations were from 1977 to 2018, since the period 1970–1976 was used for the model’s warm–up.

The Claise basin corresponds to the Brenne Natural Regional Park. The most accurate documentation for its history is the work of Benarrous (2009) that has been cited many times throughout the manuscript. Benarrous (2009) fulfilled a PhD thesis for the University of Paris I – Sorbonne that tackled the history, archaeology and paleo–environment of the Brenne Park part of which is the Claise. The earliest records for the study area (either grey or scientific literature) indicate that the whole park was an abandoned land that was avoided to be dealt with due its several water–borne diseases and relative inaccessibility. Accordingly, the whole territory was called “the bad lands” and almost no human activity took place there. As explained in the manuscript, the particular pedology created many challenges and severely limited the use of the often water–logged soils. Other than water stagnation, soils of the study area served and still serve very little. The only form of human intervention in the basin was the creation of the ponds during the middle ages. Land occupation changes since historical times were minor due to the infertility and the reduced eco–services that could be provided from the basin.

The limited agricultural activities only started to appear in the year 1914 while the landscape always consisted of grasslands, forests and ponds. The Claise and the Brenne Park then acquired the status of a Regional Park in 1989, and took part of NATURA 2000, ZNIEFF types I and II and RAMSAR conventions. Therefore, soil/land occupation activities have been strongly regulated ever since. The study area has been somewhat stable throughout history with the only changes being the proliferation of ponds. Human activities other than pond proliferation are relatively inexistent and therefore the human–effect on soil properties is also minor.

The period 1970–2018 was chosen since it presents a complete climatic cycle, the year 2008 is the year of the HWSD database. The year 2018 was chosen since it reflects the most current land occupation of the basin, therefore accounting for the maximal number of ponds present, in order to precisely determine their effect. Since the year 1970 no changes have occurred in soil properties or in the land occupation other than the ponds. In strictly regulated environments such as the Claise, changes in soil and land occupation properties do not occur dramatically in such timeframes. Therefore, in the particular case of the Claise, the temporal difference is not problematic, since climate is the only variable factor in the basin.   

The Authors would like to thank you for pointing out this issue and acknowledge that a historical clarification to justify this point is appropriate. Accordingly a footnote for Table 1 was added to explain this case.

The footnote can be found in lines 181–183: “† The correspondence of the Claise basin to a Natural park renders any human induced modifications on soils minor. Further the Claise’s land occupation pattern is relatively stable. Therefore, climate is the only variable factor in the study area. Hence, the temporal difference of the utilized datasets does not cause temporalinduced biases.”

4) Line 264: “[76, 38] have …”: Such in–text citation is inappropriate. Please revise.

Thank you for noticing this. The following modification was performed:

Line 277: “Almendinger et al. (2014) and Jalowska and Yuan (2019) [77,39], have shown that an accurate representation of basin processes requires the integration of the SWAT model’s impoundments function.”

5) Line 283: Figure 3 is not “soil erodibility”. Should be soil erodibility parameters.

The following correction was made:

Line 297: “Soil erodibility parameters are presented in Figure 3.”

6) Line 370: “replacing ponds by their surrounding cover”: by the dominant cover?

All ponds in the basin are surrounded by grasslands. With reference to Table 3, the surrounding cover (grasslands) is also the dominant cover.  For this reason the modification “by their surrounding dominant cover”, in line 386, was made.

7) Table 6: According to the results, a large part of the moderate class is misinterpreted as low class. Some explanation should be provided here.

Thank you for highlighting. As mentioned previously, the INRA maps according to their description are not entirely suitable for basin scale use. However, these are the official erosion maps used in France. These were used in order to validate the established erosion map. Differences between the moderate and low classes are due to the fact that “no–erosion” zones do not exist in the INRA maps, but are classified as low erosion classes. The discrepancy in the results is due to this fact, given the finer representation scale of the produced maps in the manuscript.

8) Line 413: “a complete shift from low to moderate risks is observed …” in?

Thank you for pointing this out. The following correction was made: “At a graphical scale, a complete shift from low to moderate risks, in the greatest part of the basin, is observed” (lines 426–427).

_____________________________________________________________________________

The Authors would like to express, once again, their gratitude to the Editor and the Reviewers for leveraging the quality of this work.

Round 2

Reviewer 2 Report

The manuscript has been substantially improved, and I have no additional comments. Thanks for the efforts. 

Author Response

1. The letters and numbers in Figures 1, 2, 3, 5, 7, 8 and 9 are too small.
The Authors thank the Editor for this remark. The following modifications were made:
Figure 1: All elements of figure 1 were resized. The font size was increased from 10 to 14, the
scale bar label was modified, and the old figure was replaced
Figure 2: The font size was increased from 11 to 14, and the old figure was replaced.
Figure 3: The legend’s font size was increased, all elements were resized, the scale bar label was
modified, and the old figure was replaced.
Figure 5: The legend’s font size was increased, all elements were resized, the scale bar label was
modified, and the old figure was replaced.
Figure 7: The legend’s font size was increased, all elements were resized, the scale bar label was
modified, and the old figure was replaced.
Figure 8: The legend’s font size was increased, all elements were resized, the scale bar label was
modified, and the old figure was replaced.
Figure 9: The legend’s font size was increased, all elements were resized, the scale bar label was
modified, and the old figure was replaced.
2. Please, check Equation (1)!
Thank you for highlighting. Equation (1) was corrected following the SWAT user manual and
was replaced by: “sed = 11.8 · (Qsurf · qpeak · areahru) 0.56 · KUSLE · CUSLE · PUSLE · LSUSLE ·
CFRG” in line 149.
Accordingly, lines 150-154 were modified to: “where: sed is the HRU sediment yield (t); Qsurf is
the surface runoff volume (mm); qpeak is the peak runoff rate (m3/s); areahru is the area of the
HRU (ha); KUSLE is the Universal Soil Loss Equation (USLE) soil erodibility factor; CUSLE is the
USLE cover and management factor; PUSLE is the USLE support practice factor; LSUSLE is the
USLE topographic factor; and CFRG is the coarse fragment factor.”
3. More details are needed for the explanation of Table 6.
Thank you for highlighting. The following modification was made: “As can be seen from Table
6, a total of 134 validation points were chosen. These were divided into 86, 37 and 11 lowerosion,
moderate-erosion, and high-erosion zones validation points, respectively. A large part
of the moderate-erosion class was misinterpreted as the low-erosion class. This discrepancy is
due to the fact that the no-erosion zones do not exist in the INRA maps, but are instead classified
as low-erosion zones. Therefore, the error margin in the moderate-erosion class from Table 6 is
justified by the finer scale representation of the produced maps, compared to the INRA maps” in
lines 409-415
4. Line 404: What is the Cohen's Kappa coefficient?
Thank you for pointing out this issue. The following modifications were made:
a) Line 415-416: The following modification was made: “The overall accuracy was determined
to be 75%, while the computed Cohen's Kappa coefficient [82]”and the complete reference was
added to the reference list
82. Cohen J. A coefficient of agreement for nominal scales. Educational and Psychological
Measurement. 1960; 20(1), 37-46.
b) Line 417-418: “The Kappa coefficient was used since it tests inter-rater reliability; i.e: the
coefficient represents the extent to which the generated data are correct representations of the
measured data.” was added
5. See the annotated manuscript!
The Authors would like to express their deepest forms of gratitude for the detailed corrections in
the annotated manuscript. Each correction was attended to and modified accordingly. The track
changes function was used as requested.
____________________________________________________________________________________
The Authors would like to express, once again, their gratitude to the Editor for leveraging the
quality of this work.
